# Cryo-electron tomography provides topological insights into mutant huntingtin exon 1 and polyQ aggregates

Jesús G. Galaz-Montoya [1✉], Sarah H. Shahmoradian[2,6], Koning Shen[3,5,6], Judith Frydman [3] & Wah Chiu [1,4✉]

Huntington disease (HD) is a neurodegenerative trinucleotide repeat disorder caused by an expanded poly-glutamine (polyQ) tract in the mutant huntingtin (mHTT) protein. The formation and topology of filamentous mHTT inclusions in the brain (hallmarks of HD implicated in neurotoxicity) remain elusive. Using cryo-electron tomography and subtomogram averaging, here we show that mHTT exon 1 and polyQ-only aggregates in vitro are structurally heterogenous and filamentous, similar to prior observations with other methods. Yet, we find filaments in both types of aggregates under ~2 nm in width, thinner than previously reported, and regions forming large sheets. In addition, our data show a prevalent subpopulation of filaments exhibiting a lumpy slab morphology in both aggregates, supportive of the polyQ core model. This provides a basis for future cryoET studies of various aggregated mHTT and polyQ constructs to improve their structure-based modeling as well as their identification in cells without fusion tags.

[1] Department of Bioengineering and James H. Clark Center, Stanford University, Stanford, CA, USA. [2] Department of Biology and Chemistry, Laboratory of Biomolecular Research, Paul Scherrer Institute, Villigen, Switzerland. [3] Department of Biology, Stanford University, Stanford, CA, USA. [4] Division of CryoEM and Bioimaging, SSRL, SLAC National Accelerator Laboratory, Menlo Park, CA, USA. [5]Present address: Department of Molecular and Cell biology, University of California, Berkeley, CA, USA. [6]These authors contributed equally: Sarah H. Shahmoradian, Koning Shen. ✉email: jgalaz@gmail.com; wahc@stanford.edu

Huntington disease (HD) is a neurodegenerative, fatal tri-nucleotide repeat disorder caused by a CAG expansion in exon 1 of the huntingtin gene (HTT) yielding a mutant protein (mHTT) with a polyQ tract exceeding a pathogenic threshold of Q > ~35[1]. HD patients suffer from severe motor and cognitive impairments and despite our increased understanding of HD[2] and promising clinical trials[3], cures and preventive treatments remain elusive[4].

Expression of mHTT exon 1 (a caspase cleavage product within cells, hereafter mEx1) elicits HD phenotypes in cellular and animal models[5–8], including primates[9]. Furthermore, mEx1 inclusions in human brains[10] are morphologically similar to those in transgenic[11] and mEx1 knock-in[12] mice.

A polyQ expansion in different genes causes at least eight other disorders with varying pathogenic Q-repeat length thresholds[13,14], and polyQ peptides as short as Q = 20 are toxic when they contain a nuclear localization signal[15]. Since structure often determines function[16], as shown for mHTT toxic aggregates[17,18], an increased structural understanding of polyQ aggregates can help uncover the mechanisms underlying their biogenesis, development, and cytotoxicity to better model polyQ disorders.

Both small mHTT oligomers and large inclusion bodies can be neurotoxic[19,20]. Filamentous aggregates of mEx1 constructs with various polyQ lengths (mEx1-Qn) have been amply visualized with negative staining transmission electron microscopy (NS-TEM)[21–24], a technique often limited to 2D projections and subject to metal stain and drying artifacts. Two recent studies used cryo-focused ion beam milling and electron tomography (cryoFIB-ET) to visualize transfected mEx1-Q97 forming inclusion bodies within yeast[25] as well as in cultured HeLa and mouse primary neuronal[26] cells. However, a green fluorescence protein (GFP) fusion tag was used in the former, which can alter mEx1 aggregation[27], and detailed analyses of filament topologies were not pursued.

Here, we used direct observation (without heavy metal stain or fusion tags) by cryo-electron tomography (cryoET) and subtomogram averaging (STA)[28,29] to visualize vitrified filamentous mEx1-Q51 and Q51 (a peptide consisting of only glutamines) aggregates in vitro. We leveraged our initial observations of mEx1-Q51 filaments by cryoET[30–32] and capitalized on recent algorithmic developments including compressed sensing for tomographic reconstruction[33,34], convolutional neural networks for feature annotation[35], and automated fiducial-less tiltseries alignment and subtiltseries refinement for STA[36] to resolve previously unattainable structures. Our study provides a three-dimensional (3D), nanometer-resolution structural description of untagged, vitrified mEx1 and Q-only aggregates, finding filaments that are thinner than previously observed, laminated sheets, and a predominant conformation exhibiting a lumpy slab morphology that supports the polyQ core model.

## Results

### Mutant huntingtin exon 1-Q51 filaments exhibit a large variation in width, narrow branching angles, and lamination.
We analyzed tomographic tiltseries of vitrified mEx1-Q51 aggregation reaction and their corresponding reconstructed tomograms (Fig. 1a) (see "Methods"). Owing to the higher contrast and minimized missing wedge artifacts attainable with compressed sensing[33] compared to standard weighted back projection, we incorporated the former method in our pipeline to reconstruct the tiltseries into tomograms (see "Methods"), which exhibited aggregated filamentous densities (Fig. 1b). While compressed sensing might introduce artifacts at high resolution in the subnanometer range, it has been demonstrated to produce faithful reconstructions at nanometer resolution[37]. Capitalizing on our

prior experience[30–32], a relatively short incubation time under our experimental conditions (Methods) ensured we would see polymerized filaments in aggregates, but not so large as to not fit in the holes of the holey-carbon cryo-electron microscopy (cryoEM) support grid or as to preclude penetration by the electron beam. Importantly, incubation was long enough to ensure cleavage of the GST tag (Supplementary Fig. 1, Supplementary Data 1), which has been shown not to be incorporated into polymerized filaments[21,38,39]. As a result, a sampling (n = 250) of filament lengths ranged between ~7.3 nm for the shortest nascent branches and protofilaments detectable in downsampled tomograms and ~654.2 nm for the longest continuous densities (mean 95.1 nm, standard deviation 106.6 nm). The most frequently observed widths from aggregates in six tomograms ranged between ~5 and ~16 nm, with the thinnest filaments exhibiting regions down to ~2 nm thickness (Fig. 1c, d). On the other hand, the thickest filaments measured over ~20 nm in width. These measurements are not consistent with a cylindrical shape of a single radius, as reported in recent cryoFIB-ET studies[25,26]. Rather, our observations are consistent with a heterogeneous plethora of thin filaments, rectangular prisms, and even sheets of varying size. We interpret the predominant species among our observed filaments as 3D rectangular slabs, which could exhibit many different center-slice widths in between their widest and narrowest dimensions when sliced computationally at slanted angles. Our computational simulations of filamentous subtomograms using EMAN2[40] support this model (Supplementary Fig. 2).

We used semi-automated annotation based on neural networks as implemented in EMAN2[35] to visualize in 3D the morphology of the objects yielding the extensive width variations detectable in 2D slices of our 3D tomograms (this was the only step in our analyses that used machine learning technology). Visualizing mEx1 filaments as 3D isosurfaces (Fig. 1b; Supplementary Fig. 3) revealed filaments of different dimensions altogether, including regions that appeared as sheets as thick as ~50 nm, estimated from the annotations and from their persistence through 2D slices. The mEx1 filaments seemed to predominantly branch out at angles varying from ~10° to ~45° (only sporadically larger), with angles between ~20° and ~25° being most common (Fig. 1e).

### Subpopulation of mEx1-Q51 filament segments exhibits a lumpy, slab-shaped morphology.
Many filaments appeared to be lumpy both in 2D slices from 3D tomograms (Fig. 2a) as well as in 3D annotations (Fig. 2b), suggestive of potential periodicity. Thus, we performed STA of manually selected filament segments, avoiding obviously laminated regions and thick bundles. The subtomogram average of mEx1-Q51 filament segments (n = 450, from six tomograms) converged to a lumpy ~7 × 15 nm slab at ~3.5 nm resolution (Fig. 2c). The Fourier transform of 2D projections of the average did not reveal crisp layer lines, in agreement with previous studies suggesting that mEx1 filaments do not exhibit a canonical amyloid structure with parallel subunits stacked helically in register[23]. Indeed, HD does not strictly fit among the diseases known as amyloidoses[41]; nonetheless, the power spectra of STA projections showed bright maxima off of the meridian, at ~11.7 nm (Supplementary Fig. 5a), suggestive of potential periodicity for at least relatively short stretches (~65 nm, the length included in the extracted subtomograms).

### Lumpy, slab-shaped Q51-only filaments also exhibit lamination.
Since an expanded polyQ tract is the common culprit of all polyglutamine diseases, we performed the same analyses for a Q51-only peptide (Fig. 3a) as reported above for mEx1-Q51. We found that Q51 also forms aggregates (Fig. 3b, Supplementary Fig. 4) exhibiting lumpy filaments (Fig. 3c) of varying lengths

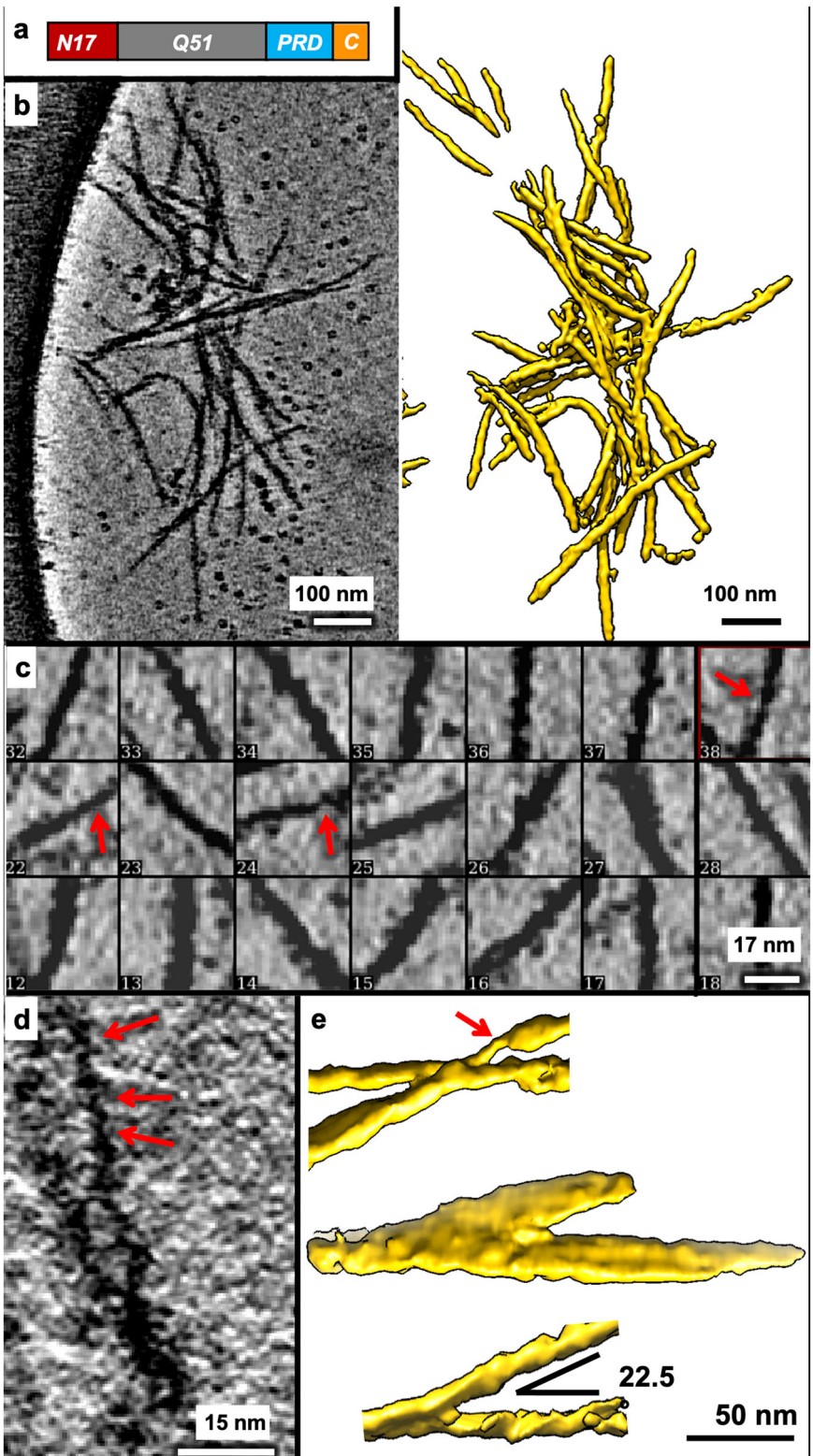

**Fig. 1 MEx1-Q51 filaments exhibit a large variation in width within and across filaments. a** Schematic of the mEx1-Q51 construct. **b** Slice parallel to xy (~1.7 nm thick) through a representative 4x downsampled cryoET tomogram of aggregated mEx1-Q51, reconstructed with compressed sensing, lightly filtered to enhance visualization, and corresponding semi-automated 3D annotation. **c** Selected areas from slices of large mEx1-Q51 aggregates showing individual filament segments, widely varying in width, with the thinnest filaments exhibiting regions down to ~2 nm width, indicated by the red arrows. **d** Zoomed-in view of a xy slice (~0.4 nm thick) from a selected region of a tomogram without any downsampling, showcasing ultra-thin regions in mEx1-Q51 filaments. **e** Sections of annotated mEX1-Q51 filamentous aggregates from cryoET tomograms showing relatively narrow branching angles and an example of a thicker laminated sheet-like region (the annotation example in the middle). Scale bars: 100 nm (**b**), 17 nm (**c**), 15 nm (**d**), 50 nm (**e**).

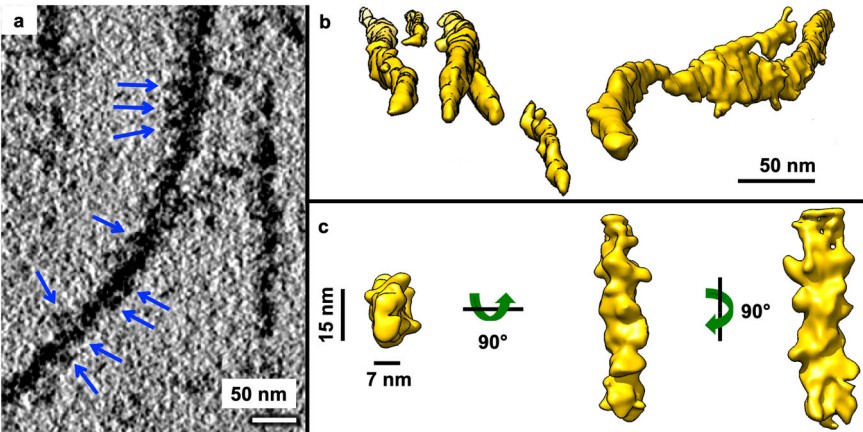

**Fig. 2 Aggregated mEx1-Q51 exhibits lumpy, slab-shaped filaments. a** Pseudo-periodic pattern of repeating lumps (blue arrows) along the length of an mEx1-Q51 filament as seen in an xy slice (4.4 Å thick) from a tomogram of aggregated mEx1-Q51. **b** Selected regions from semi-automated neural network annotations showing lumpy filaments of various widths, including sheet-like regions (middle region of right-most example). **c** Subtomogram average of a subpopulation of filament segments (n = 450) exhibiting a lumpy ~7 × 15 nm slab-shaped morphology. Scale bars: 50 nm (**a**, **b**), 15 nm and 7 nm (**c**).

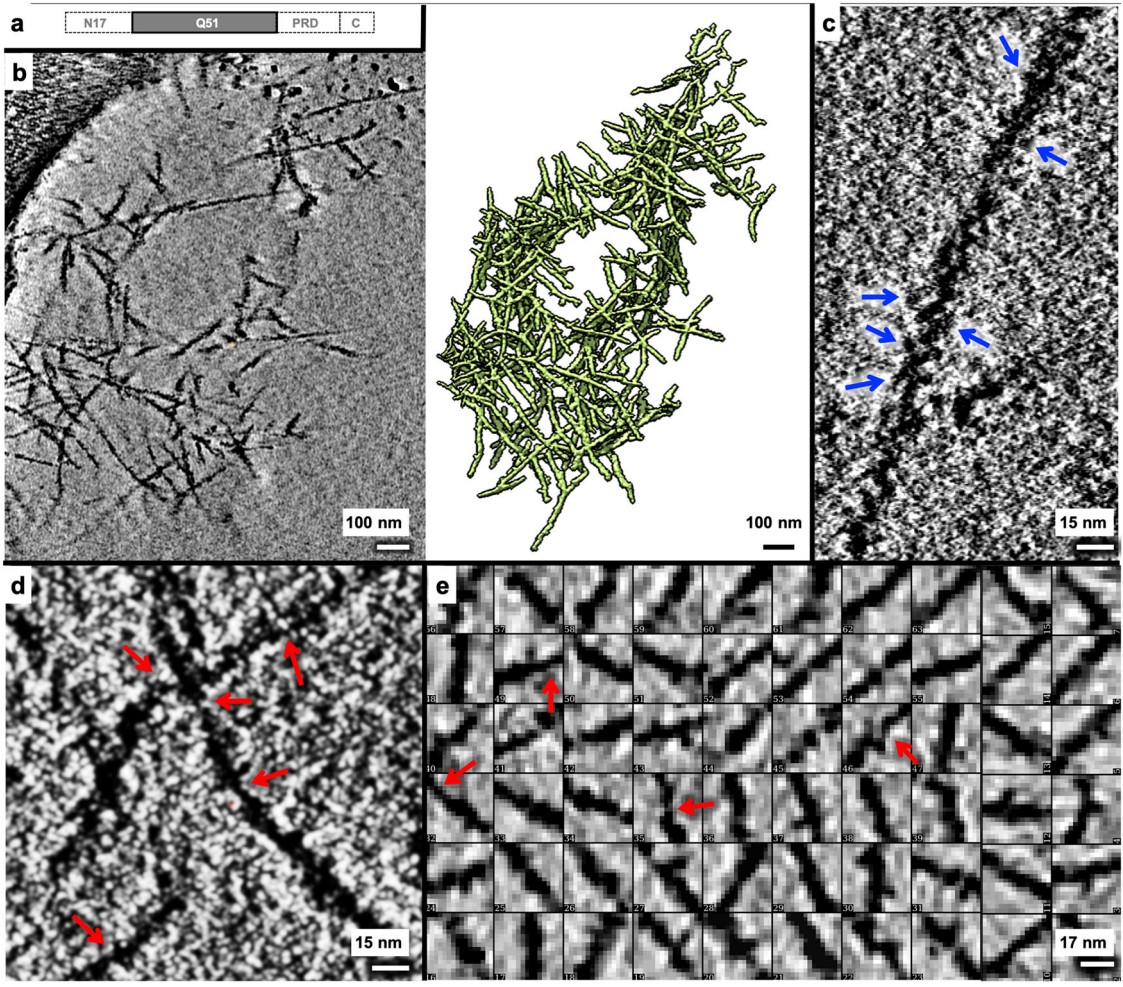

**Fig. 3 Lumpy Q51 filaments exhibit a large range of widths. a** Schematic of the Q51 construct, lacking all mEx1 domains except for the polyQ tract. **b** Slice parallel to the xy plane (~2.1 nm thick) through a representative 4x downsampled cryoET tomogram of aggregated Q51 reconstructed with compressed sensing and corresponding 3D annotation. Zoomed-in views of xy slices (~0.5 nm thick) from selected regions of the tomogram shown in "**a**" but without any downsampling, exhibiting **c** a pseudo-periodic pattern of repeating lumps along the length of a Q51 filament (blue arrows), and **d** regions in thin filaments that are as thin as ~2 nm in width (red arrows). **e** Examples of 2D xy slices through representative 3D subtomograms of Q51 filament segments showing a wide variation in width, including super-thin regions ~2 nm in width (red arrows). Scale bars: 100 nm (**b**), 15 nm (**c**, **d**), and 17 nm (**e**).

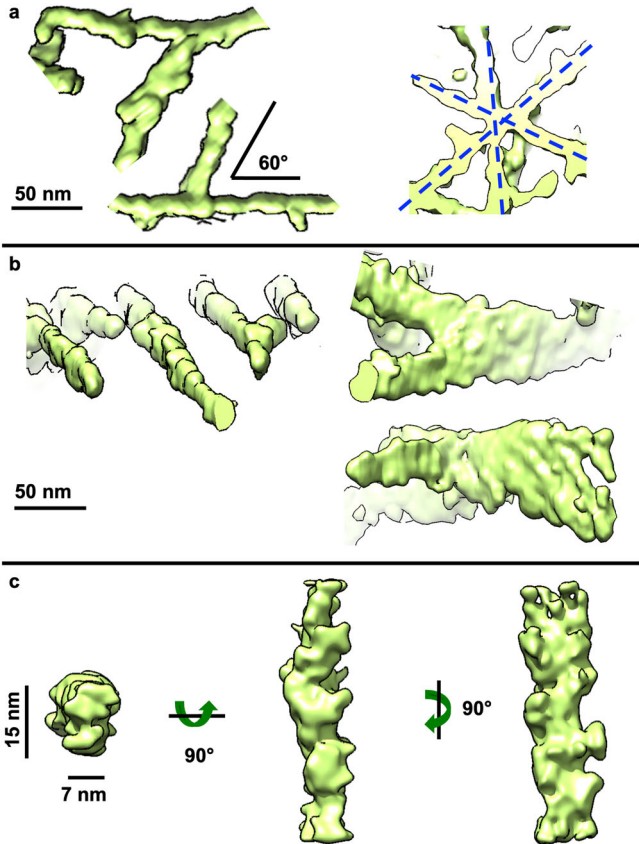

**Fig. 4 Aggregated Q51 exhibits lamination sheets and predominantly lumpy, slab-shaped filaments. a** Representative sections of annotated Q51 filamentous aggregates from cryoET tomograms showing their most common branching/crossover angle (~60°), often in an asterisk-like pattern, and **b** thicker regions akin to lamination, onside thinner ones. **c** Subtomogram average of a subpopulation of filament segments ($n = 493$) exhibiting a lumpy ~7 × 15 nm slab-shaped morphology. Scale bars: 50 nm (**a**, **b**), 15 nm, and 7 nm (**c**).

($n = 250$, 4.6–891.3 nm range, mean 98.6 nm, and standard deviation 130.4 nm) and widths (Fig. 3d, e), with regions as thin as ~2 nm.

Q51 filaments branched out/crossed over more often and at wider angles than mEx1-Q51 filaments, with ~60° being the most common angle (Fig. 4a). Furthermore, Q51 filaments exhibited larger lamination sheets than those of mEx1-Q51 of up to ~60+ nm in width (Fig. 4b). The subtomogram average of non-laminated filament segments ($n = 493$, from six tomograms) converged to ~3.2 nm resolution and also revealed a lumpy ~7 × 15 nm slab (Fig. 4c), with a crossover length of ~11.2 nm according to the power spectrum of its projections (Supplementary Fig. 5b), all strikingly similar results to those obtained for mEx1-Q51.

## Discussion

In one of the earliest reports visualizing mEx1-Q51 filamentous aggregates by NS-TEM, filaments digested with factor Xa or trypsin, which removes mEx1's N17 domain critical to mHTT localization and function[42], were reported to have a diameter of ~7.7–12 nm from two-dimensional (2D) images[21]. These were occasionally referred to as ribbons, and other 2D NS-TEM observations have reported similar filaments with a ~10–12 nm diameter, which may associate laterally[23]. However, apparent lateral associations in 2D NS-TEM observations may arise from

the compression of all densities onto a single layer due to dehydration, which is less likely to occur in thick, hydrated cryoET specimens, even though some adsorption to the air-water interface at the bottom and top surfaces of the ice cannot be entirely discarded[43]. In contrast, the vitrified filaments seen in our 3D tomograms varied much more in width within and across filaments, similar to those reported in studies visualizing aggregated Q-only peptides with NS-TEM[44,45], which also detected wide ribbons and thin filaments under different incubation temperatures and using a freeze-concentration method involving cycles of freezing and thawing, and in mEx1 studies that reported temperature-dependent polymorphs[38].

While cryoEM has been amply used to study other amyloidogenic filaments[46–53], cryoEM studies of mHTT and polyQ-containing aggregates have been scant in comparison, likely due to the extensive conformational heterogeneity of these specimens[54], which also limits the applicability of bulk techniques (e.g., circular dichroism) and calls for the increasing application of single-molecule techniques[55]. Single-molecule techniques such as atomic force microscopy (AFM) and various modalities of electron microscopy (EM) can observe individual components in aggregates (molecules, oligomers, protofilaments, filaments, and pleomorphic aggregates in non-filamentous conformations). In data from EM-related methods, such components can be classified prior to averaging. Of note, cryoET is the most suitable method to investigate the overall structure of relatively large/thick, hydrated samples exhibiting extensive conformational and compositional heterogeneity as it minimizes the confounding effect of potentially overlapping densities from different components, as in 2D projections produced by single-particle cryoEM or NS-TEM.

Two recent cryoFIB-ET studies annotated mEx1-Q97-GFP filaments in tomograms of transfected cellular systems but did not average them. The filaments were either modeled as cylinders with an 8-nm diameter during template-based annotation[26] or were segmented as 16 nm filaments[25], surprisingly twice as thick in the latter study than in the former, perhaps owing to differences in the non-native expression systems used or to the confounding presence of GFP fusion tags. Indeed, there can be caveats to using fusions to fluorescent proteins as tags, from impairing the viability and growth of cells via toxic effects from tag aggregation, excitation, or photoactivation, to changing the structure, function, and cellular localization of the tagged protein[56]. The filament homogeneity reported in these cellular studies may arise from differences in the constructs and experimental conditions used or the biochemical context (indeed, mEx1 aggregates are polymorphic[39]); however, template bias may have also contributed to the interpreted and reported homogeneity. While template-based approaches have been successfully applied to annotate more regularly shaped biological filaments[57,58], our data here suggest that the use of a cylindrical template is not an adequate approach to annotate widely heterogeneous mEx1 and polyQ aggregates with filamentous densities of varying dimensions. When identifying features in tomograms, template matching can be biased[59] and manual human annotation is subjective and therefore often uncertain and inconsistent[60]. Here, we used template-free, semi-automated annotation based on machine learning since it can ameliorate these issues by minimizing human input and the use of a priori constraints inherent in templates[35].

A study by AFM on mEx1-Q49 aggregation suggested that nucleated branching from filaments, rather than lateral associations among them, leads to large bundles[61]. However, branching does not explain how the thinner filaments that we observe here (under 7 nm thick, down to ~2 nm thickness) would assemble into thicker slabs and sheets without associating laterally or

growing transversally to the main filament axis. Rather, our results suggest that preformed thin filaments, for both mEx1-Q51 and Q51, can associate laterally and/or that growing filaments can expand transversally in addition to longitudinally, akin to the lamination observed for Aβ[62]. These complementary mechanisms may be at play in parallel to nucleated branching or at different stages of aggregation. Importantly, the ability of mEx1 proto-filaments to associate laterally has also been suggested by recent NS-TEM experiments that observed striations in wide filaments[39].

While AFM surface measurements are limited to ~30 nm in lateral resolution and are taken from specimens that are often absorbed and dried onto a 2D substrate, an earlier AFM study of aggregated Q44 peptide detected regions in filament tips with a height (the size of the specimen in the direction perpendicular to the adsorption surface) as thin as ~5 nm[63]. This and the thinness of some of the filament regions we observed here (as thin as ~2 nm) seem to disagree with the minimum width of ~7–8 nm proposed for polyQ filaments from various Qn constructs in a prior nuclear magnetic resonance study that also presented NS-TEM images[64]. On the other hand, the latter study also reported filament widths for a Q54 peptide from NS-TEM images from ~7–8 nm up to ~16 nm, in striking agreement with the short and long sides of the slab-shaped model we propose here as the predominant morphology for filaments formed by both mEx1-Q51 and Q51.

The morphological characteristics deviating from thin cylindrical shapes to form lumpy slabs and sheets may serve as structural hallmarks to identify untagged mHTT aggregates in cells. Furthermore, the more frequent and wider-angle branching of Q-only filaments compared to mEx1 is consistent with our prior 2D cryoEM observations[32], suggesting that the N17 domain promotes inter-filament bundling. This difference emphasizes the role of the domains flanking the polyQ tract in dictating overall aggregate morphology. Conversely, the occurrence of branching might be primarily polyQ-driven.

Our 3D observations here complement and clarify results from previous studies by NS-TEM and AFM, as well as light microscopy[65,66], which visualized mEx1 filamentous aggregates at a coarser level: features often described as globules or oligomers or thick filaments actually correspond to bundles of many interwoven thinner filamentous densities when viewed by cryoET.

The fact that in spite of overall aggregate differences the predominant populations for both mEx1 and Q-only filaments exhibit a similar lumpy-slab shape and distance between putative crossovers as revealed by STA suggests that the morphology of their core is dictated by and primarily comprised of the polyQ tract, and that the flanking domains in mEx1 are largely exposed at the filament surface, allowing them to modulate inter-filament aggregation. This interpretation agrees with previous nuclear magnetic resonance studies on non-pathogenic[67] and pathogenic[68,69] mEx1 variants that propose the existence of a dense polyQ core. Interestingly, one of the most recent studies supporting this model[39] suggests that the absence of the N17 domain from mEx1 constructs yields filaments that are smoother, while the presence of N17 makes filaments fuzzier. This seems to agree visually with our subtomogram averages and may explain the slightly lower resolution we obtained for the average of mEx1-Q51 filament segments compared to polyQ alone.

In one of the latest studies supporting the polyQ-core model[38], the authors observed mEx1-Q44 filaments formed at two different temperatures by 2D NS-TEM images (presented in the supplement). The widths reported for these filaments were ~6.5 nm and ~15.2 nm, in striking agreement with the dimensions of our slab-shaped subtomogram averages of filament segments from 3D cryoET tomograms of vitrified mEX-Q51 and Q51. While their

hypothesis that the thicker ~15.2 nm filaments must arise from the association of two protofilaments ~6.5 nm thick and their updated model for this interaction[39] seem to be somewhat compatible with our observations here, their proposal that the flanking domains mediate such association does not explain our observation that polyQ-only filaments also yield a dominant subpopulation with the same ~7 × 15 nm slab morphology, which could also correspond to two associated protofilaments without flanking domains to bind them. Furthermore, their model does not account for possible twisting, which may underly the pseudo-periodicity we observe here. If, indeed, our mEx1-Q51 and Q51-only predominant subpopulations of ~7 × 15 nm filaments are composed of two thinner interwinding protofilaments, the data suggest that they might be bound primarily via polyQ-polyQ interactions.

Our observations here warrant further cryoET experiments with much larger datasets of aggregation-competent mEx1 and polyQ-only constructs devoid of solubilization and purification tags, as even these can cause modest alterations in aggregation kinetics[70,71]. CryoEM/ET datasets at higher magnification and contrast, using state-of-the-art instrumentation, could test whether there exist filament species even thinner than the ~2 nm regions we observed here, and could probe the effects of increasing polyQ length on the 3D morphologies of vitrified filamentous aggregates. Longer incubation times would yield larger aggregates likely containing much longer filaments than measured here, but the bulk thickness of the aggregates would preclude visualization by cryoET. However, this limitation could be overcome by the use of focused ion beam milling to render the specimens amenable to cryoET[72]. Assessing the effects of post-translational modifications on filament and overall aggregate structures with cryoET might be particularly interesting as some of these modifications modulate mEx1 aggregation with neuroprotective effects[73]. Finally, sonication concomitant with trypsin digestion of mEx1 filaments might yield a homogenous-enough population of the polyQ core that may be more amenable to higher-resolution cryoEM/ET studies.

## Methods

**In vitro mEx1-Q51 and Q51 peptide aggregation assays and cryoET sample preparation**. We used a mutant huntingtin (mHTT) exon 1 with 51 glutamine repeats (mEx1-Q51) and a polyQ-only peptide with 51 repeats (Q51), each of them fused to a TEV cleavage sequence and a GST tag, as also used in our previous study[32]. The mEx1-Q51 and Q51 constructs were expressed in a BL21(DE3) bacteria (Agilent) strain. Bacterial pellets expressing a pGEX-mHTT-Ex1-Q51 plasmid (or a PolyQ-only variant) were resuspended in 50 mM sodium phosphate, pH 8.0; 150 mM NaCl; 1 mM EDTA and lysed. Lysate was then incubated with GSH-Sepharose resin (GE Healthcare) and washed with 0.1% Triton, 500 mM NaCl, and 5 mM Mg-ATP. Protein was eluted with 15 mM Glutathione. Protein was then concentrated and buffer exchanged into 50 mM Tris-HCl, pH 8.0; 100 mM NaCl; 5% glycerol. Concentrated protein was 0.2 µm filtered before storage at −80 °C to remove any protein aggregates that occurred during the purification process. Aggregation was initiated separately at a concentration of 6 µM for each construct in vitro by the addition of AcTEV™ protease (Invitrogen), as previously described for mEx1-Q51[30]. The samples were incubated at 30 °C before vitrification. Aliquots of 2.5 µm were separately applied to 200-mesh holey carbon Quantifoil copper grids (previously washed with acetone and rinsed in PBS) between 4 and 6 h post-initiation of aggregation. The grids were plunge-frozen in a liquid ethane bath kept at liquid nitrogen temperature using a Vitrobot Mark III (FEI Instruments).

**ThioflavinT aggregation assay and AcTEV cleavage assay**. For the ThioflavinT aggregation assay, aggregation reaction was prepared as above and combined with 12.5 µM ThioflavinT dye (Sigma-Aldrich). Reactions were transferred to a 3904 Corning plate and read with an Infinite M1000 plate reader (Tecan Systems). Plate reader conditions were 30 °C incubation, 446 nm excitation, 490 nm emission, reading every 15 min. To assess the efficiency of AcTEV cleavage, aggregation reaction was prepared as above, and timepoints were taken by combining the aggregation reaction with 4x Laemmli sample buffer, boiling at 95 °C for 5 min, and storing at −20 °C until all timepoints were collected. Then, timepoints were run on a 12% SDS-PAGE gel and stained with Coomassie for imaging.

**Table 1 Cryo-EM data collection, refinement and validation statistics.**

|  | mEx1-Q51 (EMDB-21248) | Q51 (EMDB-21253) |
|---|---|---|
| *Data collection and processing*[a] | | |
| Magnification | 25 K | 20 K |
| Voltage (kV) | 200 kV | 200 kV |
| Electron exposure (e–/Å²) | 62 e/Å² | 80 e/Å² |
| Defocus range (μm) | 5 μm | 6 μm |
| Pixel size (Å) | 4.4 Å/pixel | 5.29 Å/pixel |
| Symmetry imposed | C1 | C1 |
| Initial particle images (no.) | 450 | 493 |
| Final particle images (no.) | 225 | 247 |
| Map resolution (Å) | 35 Å | 32 Å |
| FSC threshold | 0.143 | 0.143 |

Model refinement is not applicable to low-resolution cryoET structures.
[a]See "Methods" for further details.

**Tiltseries collection**. We collected six tiltseries of the Q51 peptide using SerialEM software in low-dose mode[74] on a JEM2100 electron microscope operated at 200 kV and equipped with a CCD camera and a LaB6 electron source, from −60° to 60° in 2° increments, at 6 μm target underfocus, 5.29 Å/pixel sampling size, with a cumulative dose of ∼80 e/Å² (Table 1). We also reanalyzed a previous dataset comprised of 20 tiltseries of mEx1-Q51 + TRiC[30], collected similarly to the Q51 peptide dataset, using the same electron microscope, electron source, and recording device, but with a slightly finer sampling size of 4.4 Å/pixel, 5 μm target underfocus, and ∼62 e/Å² cumulative dose.

**Tomographic reconstruction**. All mEx1-Q51 and Q51 tilt series were binned by 2x and initially aligned and reconstructed into tomograms with IMOD[75]. Images with artifacts (grid bars in the field of view blocking large regions of the specimen, evident large drift, obvious radiation damage, etc.) were manually removed prior to tiltseries alignment and tomographic reconstruction with weighted back projection. After assessing sample thickness, the tiltseries were reconstructed again using compressed sensing as implemented in ICON-GPU[33,34] to obtain tomograms with improved contrast. Of note, compressed sensing also partially restores information that is lacking due to the missing wedge artifact inherent in all single-axis limited-angle tomography experiments, such as conventional cryoET[76]. The tiltseries were aligned and reconstructed yet a third time for STA purposes (as described below), using the latest pipeline for cryoET in EMAN2[36] that performs sub-tiltseries refinement, akin to prior hybrid methods combining concepts from single-particle analysis cryoEM and STA[77–79]. We processed the mEx1-Q51 and Q51 datasets separately in virtually identical ways.

**Tomogram annotation**. Since the ultimate goal of the new EMAN2 cryoET pipeline[36] is to perform subtiltseries refinement for STA, tomogram quality only needs to be sufficient to allow for particle identification. Indeed, in EMAN2 not as many parameters are refined during tomographic reconstruction as compared to IMOD, often resulting in lower-quality tomograms. For this reason, we performed all tomographic annotations on better-quality tomograms aligned with IMOD and reconstructed with compressed sensing, as described above. MEx1-Q51 and Q51 annotations were carried out on binned-by-4 tomograms using EMAN2's neural network semi-automated annotation tools[35], except that ∼2–3x as many references as the 10 recommended were segmented to seed annotation, and ∼2–3x as many negative samples as the 100 recommended were selected to minimize false positives. We initially performed annotation of all mEx1-Q51 and Q51 tomograms by applying the convolutional neural network from the best tomogram to all the rest, separately for each specimen. However, false positives (such as annotating the carbon-hole edge and/or gold fiducials) were reduced further when we generated a neural network specific for each mEx1-Q51 and Q51 tomogram.

**Fibril width and length range measurements**. In all limited-angle tomography experiments (when you cannot tilt through the entire full range from 0° to 180° or −90° to +90° to collect a full set of projections around the object of interest), the missing wedge artifact worsens the resolution of raw tomograms along the z axis compared to that in the x and y directions, often giving the appearance of elongation of features along the axis with lowest resolution. Therefore, filament widths cannot be accurately measured in 3D from raw tomograms nor their corresponding annotations in arbitrary orientations. The most conservative measurements in the absence of averaging should be performed on slices along the z axis of reconstructed tomograms (i.e., on sections parallel to the xy plane) since features are much less well-resolved in the xz and yz planes. Here, we boxed out filament segments for STA (below) and manually measured the thinnest and thickest parts

of segments (N ∼100) from the central xy slice of the corresponding subtomogram. The mEx1-Q51 and Q51 data were separately processed in identical ways. To estimate the distribution of filament lengths, we reprojected the tomographic annotations of all the tomograms since this provided images with reduced background and without missing wedge artifacts. We manually quantified filament lengths using the Measure tool in EMAN2's e2display.py, breaking up the filaments into segments when curvature was present and adding the segment lengths up at the end, and used the Draw tool to color over measured filaments to avoid repeat measurements of the same filament in the complex, aggregated tangles.

**Initial model generation for STA**. To carry out sub-tiltseries refinement, the latest EMAN2 cryoET pipeline[36] requires that all steps (from initial tomographic reconstruction) be performed in EMAN2. However, as explained above, whole-tiltseries alignment with IMOD is often superior in quality, given its refinement of more reconstruction parameters, and reconstruction with ICON-GPU can yield higher-contrast tomograms with minimized missing wedge artifacts. Therefore, to generate an initial model, we manually extracted filament segments without much overlap from the best IMOD-aligned, compressed sensing-reconstructed tomogram for each specimen (n = 97 for mEx1-Q51; n = 135 for Q51), avoiding branching points and regions of dense bundling or obvious lamination. Then, we aligned these subtomograms to a cylindrical reference with a soft edge and computed the average using the legacy tools for STA in EMAN2[80]. This average of vertically aligned filaments was then refined constraining the angular search in altitude to only allow for slightly tilted orientations (since all particles were already vertically pre-aligned to a cylinder) and flips of 180° in altitude (the other two Euler angles were completely unconstrained). Alignment converged in ∼4–5 iterations for both datasets. We used these de novo preliminary averages as initial models for subsequent, completely unconstrained gold-standard STA of mEx1-Q51 and Q51 with sub-tiltseries refinement in the new EMAN2 pipeline.

**Subtomogram averaging**. Since the reconstruction geometry is different for tomograms produced with different software packages, we had to pick sub-tomograms of filament segments (with <∼50% overlap) manually from scratch (n = 450 for mEx1-Q51; n = 493 for Q51) in EMAN2-reconstructed tomograms. Gold-standard refinements seeded with the initial models described above converged in ∼4–5 iteration and 50% of the best-correlating particles were kept in the final average for each dataset. The subtiltseries refinement step alone improved the resolution drastically by ∼10 Å or more for both datasets, yielding averages at ∼3.5 nm and ∼3.2 nm resolution for mEx1-Q51 and Q51, respectively, according to the gold-standard FSC = 0.143 criterion.

**Visualization**. Tomographic slices were visualized with either EMAN2[40] or IMOD[75]. All isosurfaces were visualized with UCSF Chimera[81].

**Statistics and reproducibility**. We collected six tomograms of each mEX1-Q51 and Q51 aggregates and boxed out 450 and 493 subtomograms of filament segments from each set, respectively, used for filament width measurements and STA analyses, which are thoroughly described in the Methods.

**Reporting summary**. Further information on research design is available in the Nature Research Reporting Summary linked to this article.

## Data availability

The raw data can be made accessible upon request. The Electron Microscopy Data Bank accession numbers for the structures reported in this paper are as follows: mEx1-Q51 subtomogram average, EMD-21248; Q51 subtomogram average, EMD-21253.

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

## Acknowledgements

This research has been supported by grants from the National Institutes of Health, USA, No. NS092525 to J.F. and W.C., and No. P41GM103832 to W.C.

## Author contributions

All authors planned and designed experiments. K.S. generated and purified the mEx1-Q51 and Q51 constructs and ran the ThT control experiments. S.H.S. optimized and performed sample preparation plunge-freezing for cryoET, and collected the cryoET tiltseries. J.G.G.M. performed all cryoET data processing and analyses and wrote the manuscript with feedback from all authors. J.F. and W.C. provided supervision and procured funding.

## Competing interests

The authors declare no competing interests.
