## [Peer Review File · Communications Biology]

Reviewers' comments:

Reviewer #1 (Remarks to the Author):

In this work, the authors report results from their CryoET reconstructions of models for aggregates formed by a mutant, exon1 spanning region of Htt with 51 Q residues in the tract and a comparative analysis of a Q51 stretch. The authors prepare their samples sans fusion proteins. The models that emerge indicate that both constructs form heterogeneous lumpy slab like structures that associate laterally. The key results are the heterogeneity, the thinness of the filaments, clear evidence for branching and / or lateral associations. Overall, this is an easy to read MS and the data are presented in an unambiguous manner. The report is timely and will garner attention as efforts intensify to transfer the insights gleaned from these structural studies to implications in vivo. There are a few minor points that need revision / clarification and these are as follows:

1. The statement that the polyQ length threshold is the same for all known CAG repeat expansion disorders is incorrect. Please see the systematic assessments summarized by E.O. Walker in 2008 showing clear context-dependent effects on the polyQ length threshold. Additionally, the effects of flanking sequences are well established, so this statement should be corrected.
2. The narrative strives to push for convergence between the structures of Q51 and mEX1, when in fact there are visible differences and quantitative ones as well. It appears that this push on the narrative is done to convey the inference that polyQ forms the core of the filaments. That inference probably stands without insisting that the structures formed by the two constructs are very similar on all length scales. What one perceives is rather a dense mesh of entangled structures for Q51, which is concordant with previous observations.
3. The authors present a case against nucleated branching, favoring instead a model of lateral association of pre-formed filaments. The angles they observe, with a statistical preference, suggests that it is premature to make any conclusions based on the structural data. These data tells us about end products, and not about mechanisms. One would be inclined to caution against strong extrapolations because the current data are likely compatible with both models because they suggest features that go both ways.
4. The filaments are rather short, at least the ones for which we are shown pictures. The length distribution of filaments would be an important quantity to include.
5. Finally, there is some confusion, at least for this reader, about the manual curation vs. automation via "machine learning". The methods need clarification and precisely what form of machine learning is being used would be useful to know.

Reviewer #2 (Remarks to the Author):

This descriptive study, using cryo electron tomography, adds a more native-like view of in vitro assemblies of a model misfolding disease protein to the literature on such filamentous assemblies. The eventual relevance of the structures described here to Huntington's disease is unclear, but it illustrates that they are more diverse and more irregular than had been generally assumed. Cryo ET appears to be the best approach to describe their features, although it does not avoid surface adsorption as claimed - cryo EM samples typically adhere to the air-water interface.

The use of advanced image processing and sub tomogram averaging methods appears to be somewhat compromised by the hardware used for data collection. The type of detector is not specified (which it should be), but this seems to be the same as in a 2013 study from some of the same authors which employed a 200 kV microscope with a LaB6 source, presumably with a CCD

detector. It is not obviously a sensible strategy to push hard on the image processing with data quality far below the current state of the art.

Minor issues with the wording need to be fixed, e.g. line 274 "transforming complement and clarification", and various instances of poor language or typos.

Reviewer #3 (Remarks to the Author):

The authors present a cryo-electron tomography (cryo-ET) study of polyQ fibrils. Two polyQ variants, mEx1-Q51 and Q51, were purified from *E. coli* as GST fusion proteins and aggregation was induced by cleaving the GST tag with TEV protease. The resulting aggregates were vitrified by plunge freezing. Subsequently tomograms of fibrils were acquired and the fibrils averaged along a cylindrical reference in EMAN2. The authors find a variety of different fibril diameters (2 to 20 nm), in contrast to a recent study that described polyQ fibrils of a homogenous diameter of 8 nm in Htt inclusion bodies from HeLa cells and neurons (ref. 25). The observation of new fibrillar forms of polyQ protein is interesting, although the reasons for the discrepancy to other studies remains unclear.

Major points:

1. The authors should discuss the possibility that the different conditions in which the fibrils were produced (in vitro after TEV cleavage and in situ upon expression in mammalian cells) may be responsible for the observed difference in fibril morphology.
2. The notion that tags like GFP can alter fibril formation seems somewhat overstated. Specifically, ref. 25 also analyzed (as a control) polyQ protein without GFP tag. The statement that ref. 25 analyzed exclusively GFP-tagged polyQ protein is incorrect.
3. A major concern with the present study is that the way in which the aggregates were produced in vitro (by TEV cleavage of the GST tag) may be responsible for the heterogeneous nature of the fibrils. The TEV cleavage reaction should be shown to be complete by SDS-PAGE analysis. More importantly, the TEV cleavage reaction takes time during which cleaved polyQ protein may already begin to aggregate, possibly recruiting the polyQ tract of not yet cleaved polyQ-GST into the aggregates. Such co-aggregation is well documented in the literature and could change the morphology of the fibrils formed. Authors should perform controls to exclude the presence of non-cleaved fusion protein in the aggregates analyzed. The purification strategy should be described in more detail and the homogeneity of the analyzed proteins demonstrated. This should also include a time course of the aggregation reaction followed by ThT binding.

Reviewers' comments:

Reviewer #1 (Remarks to the Author):

In this work, the authors report results from their CryoET reconstructions of models for aggregates formed by a mutant, exon1 spanning region of Htt with 51 Q residues in the tract and a comparative analysis of a Q51 stretch. The authors prepare their samples sans fusion proteins. The models that emerge indicate that both constructs form heterogeneous lumpy slab like structures that associate laterally. The key results are the heterogeneity, the thinness of the filaments, clear evidence for branching and / or lateral associations. Overall, this is an easy to read MS and the data are presented in an unambiguous manner. The report is timely and will garner attention as efforts intensify to transfer the insights gleaned from these structural studies to implications in vivo. There are a few minor points that need revision / clarification and these are as follows:

R1.1 The statement that the polyQ length threshold is the same for all known CAG repeat expansion disorders is incorrect. Please see the systematic assessments summarized by E.O. Walker in 2008 showing clear context-dependent effects on the polyQ length threshold. Additionally, the effects of flanking sequences are well established, so this statement should be corrected.

We thank the reviewer for pointing out that our sentence introducing pathogenic polyQ expansions can be misinterpreted, and for referring us to Walker's study.

We have rephrased our sentence to be more specific and clear, as follows:

"A polyQ expansion in different genes causes at least eight other disorders with varying pathogenic Q-repeat length thresholds" (page 3, line 54).

R1.2 The narrative strives to push for convergence between the structures of Q51 and mEX1, when in fact there are visible differences and quantitative ones as well. It appears that this push on the narrative is done to convey the inference that polyQ forms the core of the filaments. That inference probably stands without insisting that the structures formed by the two constructs are very similar on all length scales. What one perceives is rather a dense mesh of entangled structures for Q51, which is concordant with previous observations.

We agree that even though our data are consistent with the polyQ-core hypothesis, the mEx1-Q1 and Q51 aggregates are different, as well as the subtomogram averages of corresponding filament segments. In the Discussion, we now re-emphasize these differences (page 14, lines 288-290, page 15, lines 296, 303-308).

R1.3. The authors present a case against nucleated branching, favoring instead a model

of lateral association of pre-formed filaments. The angles they observe, with a statistical preference, suggests that it is premature to make any conclusions based on the structural data. These data tells us about end products, and not about mechanisms. One would be inclined to caution against strong extrapolations because the current data are likely compatible with both models because they suggest features that go both ways.

We agree with the reviewer. Our intention was not to discount the nucleated branching mechanism, but rather to highlight that complementary mechanisms of aggregation (lateral association of protofibrils) or polymerization (growth in directions normal to the filament axis) may be at play and are supported by our data. We agree that our data are compatible with both models, and the different mechanisms of aggregation need not be mutually exclusive. We have refined this in the Discussion to make our point more clear, and added additional references that support this observation, such as Boatz et al., 2020, (ref. 39) from P.C.A van der Wel's group (page 13 & 14, lines 269-272).

R1.4. The filaments are rather short, at least the ones for which we are shown pictures. The length distribution of filaments would be an important quantity to include.

Even for the “small” aggregates in our present study, as required for the electron beam to penetrate through the specimen, it is challenging to objectively measure individual filament length among a tangle of filaments that curve, branch, crossover each other, coalesce, etc.

Nonetheless, in spite of the caveats, we agree that length is an important morphological parameter and thus we now provide our best estimates through descriptive statistics (maximum and minimum or “range”, mean, and standard deviation) for the distributions of filament lengths for both specimens in the Results section (page 5, lines 97-100; page 9, lines 170 & 171).

We are pursuing studies of larger aggregates that we plan to mill with a focused ion beam and expect that we might see longer filaments in such specimens.

R1.5. Finally, there is some confusion, at least for this reader, about the manual curation vs. automation via "machine learning". The methods need clarification and precisely what form of machine learning is being used would be useful to know.

The image processing software called EMAN2 uses neural-networks exclusively for tomographic annotation of the overall aggregate (i.e., to “color” the big tangle of filaments and distinguish what belongs to the tangle and what doesn't, so it can be visualized without the confounding background and noise). This semi-automated annotation method in EMAN2 that uses neural networks is documented in detail in the Nature Methods paper by Chen et al., 2017 (ref. 35). This was the only step for which we used machine learning-based technology in our study.

On the other hand, short filament segments were picked manually for subtomogram averaging. We have further clarified this in the Methods (page 7, lines 135, 136 & 146).

Reviewer #2 (Remarks to the Author):

R2.1. This descriptive study, using cryo electron tomography, adds a more native-like view of *in vitro* assemblies of a model misfolding disease protein to the literature on such filamentous assemblies. The eventual relevance of the structures described here to Huntington's disease is unclear, but it illustrates that they are more diverse and more irregular than had been generally assumed. Cryo ET appears to be the best approach to describe their features, although it does not avoid surface adsorption as claimed - cryo EM samples typically adhere to the air-water interface.

We appreciate the reviewer's comments and agree that cryogenic electron microscopy does not completely preclude surface adsorption as it is susceptible to adsorption onto the air-water interface. However, the thickness of our filamentous aggregates, with multiple layers along-z, suggest that even though adsorption may have occurred at the edges, the bulk of the specimen is hydrated. To be more conservative following the reviewer's observation, we have eliminated the portion of a sentence in our manuscript that suggested there was no surface-adsorption (or flattening) in our specimen.

R2.2. The use of advanced image processing and sub tomogram averaging methods appears to be somewhat compromised by the hardware used for data collection. The type of detector is not specified (which it should be), but this seems to be the same as in a 2013 study from some of the same authors which employed a 200 kV microscope with a LaB6 source, presumably with a CCD detector. It is not obviously a sensible strategy to push hard on the image processing with data quality far below the current state of the art.

The reviewer is correct that the Q51 data were collected with the same hardware as the mEx1-Q51 from our 2013 study. We have added the detector and electron source details explicitly in the Methods section now (page 18, line 274). While the resolution of 200 kV electron microscopes is lower than that of their 300 kV counterparts, 200 kV can provide higher contrast so long as the beam can get through the specimen and this voltage is still used in present day for routine structure determination of *in vitro* specimens.

Similarly, the first subnanometer resolution structures from digitally-recorded images were obtained with CCD cameras, demonstrating that while CCD cameras deliver lower resolution images than direct electron detectors, they become a limiting factor mostly only at very high resolutions.

While the contrast yielded by CCDs is lower compared to modern direct electron detectors, the higher dose allocated to tomographic datasets partially compensates for

this, as well as the modern Compressed Sensing reconstruction method we used here. Indeed, our tomograms show very good contrast. Rather, our subtomogram averages are limited in resolution due to specimen heterogeneity and “particle” numbers (number of segments amenable to subtomogram averaging), not by the hardware used.

Certainly, as we propose in the Discussion, we agree with the reviewer that our results here warrant further studies with much larger datasets and modern hardware, which we intend to pursue. However, our results here are not limited by the resolving power of the instrument. Indeed, a focus of this paper is to demonstrate that the use of more advanced data processing methodologies can extract new information not reported in our previous publications even from datasets collected with modest instrumentation.

R2.3. Minor issues with the wording need to be fixed, e.g., line 274 "transforming complement and clarification", and various instances of poor language or typos.

We agree that “provide a transforming complement and (provide) clarification to other studies” could be better phrased. Thus, we have changed our sentence from:

“Our observations... provide a transforming complement and clarification to previous studies by NS-TEM and AFM, as well as light microscopy”

to:

“Our observations... complement and clarify results from previous studies by NS-TEM and AFM, as well as light microscopy” (now page 14, line 291).

Reviewer #3 (Remarks to the Author):

R3.0. The authors present a cryo-electron tomography (cryo-ET) study of polyQ fibrils. Two polyQ variants, mEx1-Q51 and Q51, were purified from *E. coli* as GST fusion proteins and aggregation was induced by cleaving the GST tag with TEV protease. The resulting aggregates were vitrified by plunge freezing. Subsequently tomograms of fibrils were acquired and the fibrils averaged along a cylindrical reference in EMAN2. The authors find a variety of different fibril diameters (2 to 20 nm), in contrast to a recent study that described polyQ fibrils of a homogenous diameter of 8 nm in Htt inclusion bodies from HeLa cells and neurons (ref. 25). The observation of new fibrillar forms of polyQ protein is interesting, although the reasons for the discrepancy to other studies remains unclear.

We would like to clarify that we used a featureless cylindrical reference in EMAN2 only to obtain an “initial model” for subtomogram averaging with reduced computational time. Full subtomogram averaging refinement was then conducted following gold-standard iterative refinement without any constraints (no cylindrical constraints).

We have further clarified this in the Methods (page 21, lines 447). We have also taken the reviewer's suggestion to elaborate on the reasons for the discrepancies between different filament morphologies in different studies and now cite more references showing that huntingtin and polyglutamine filaments in general are very polymorphic and known to vary widely depending on polyQ tract length and biochemical context (see Lin et al., 2017, ref. 38; and new reference Boatz et al., 2020, ref. 39).

Major points:

R3.1. The authors should discuss the possibility that the different conditions in which the fibrils were produced (in vitro after TEV cleavage and in situ upon expression in mammalian cells) may be responsible for the observed difference in fibril morphology.

We thank the reviewer for pointing this out and have expanded on this in the Discussion (page 12 & 13, lines 248-252).

R3.2. The notion that tags like GFP can alter fibril formation seems somewhat overstated. Specifically, ref. 25 also analyzed (as a control) polyQ protein without GFP tag. The statement that ref. 25 analyzed exclusively GFP-tagged polyQ protein is incorrect.

We agree with the reviewer and have corrected the sentence making reference to this in the Introduction (pages 3 & 4, lines 64-67).

R3.3. A major concern with the present study is that the way in which the aggregates were produced in vitro (by TEV cleavage of the GST tag) may be responsible for the heterogeneous nature of the fibrils. The TEV cleavage reaction should be shown to be complete by SDS-PAGE analysis. More importantly, the TEV cleavage reaction takes time during which cleaved polyQ protein may already begin to aggregate, possibly recruiting the polyQ tract of not yet cleaved polyQ-GST into the aggregates. Such co-aggregation is well documented in the literature and could change the morphology of the fibrils formed. Authors should perform controls to exclude the presence of non-cleaved fusion protein in the aggregates analyzed. The purification strategy should be described in more detail and the homogeneity of the analyzed proteins demonstrated. This should also include a time course of the aggregation reaction followed by ThT binding.

We have added more details about the protein purification in the Methods section (page 18, lines 362-371) as well as a supplementary figure (new Figure S1) showing a time course of GST cleavage from the mEx1-Q51 construct upon addition of AcTEV protease. The reviewer can see from the Coomassie staining of the uncleaved protein in panel A that our protein purification is very clean and there is only a homogenous population of uncleaved construct at the start our aggregation reaction, and that the majority of GST is cleaved within 2 h. In panel B, corresponding ThioflavinT aggregation curves using those cleavage conditions show that the lag phase of aggregation extends well beyond 2 h, at which point most or all GST is already cleaved (much sooner than our 4 h incubation time). This indicates that the vast majority of the aggregation nucleation reaction occurs after GST cleavage and thus we can be confident that the aggregates we

analyzed contain mostly the cleaved mEx1 construct. The Q-only construct aggregates even more slowly than mEx1-Q51, as we have shown previously (Shen et al., eLife 2016; ref. 32), further arguing against uncleaved constructs contributing to the aggregation reaction.

To elaborate on this point, Scherzinger 1997 (ref. 21) suggests that uncleaved monomer does not aggregate by itself, but incomplete removal can form “globular” amorphous clumps, which are very easy to distinguish from filaments. Lastly, Boatz et al. 2020 (ref. 39) showed that the fusion tag is completely cleaved from a similar construct, MBP-mEx1-Q44, in 15 min to 1 h, supporting prior studies that showed that the uncleaved fusion protein is not incorporated into polymerized filaments (Lin et al., 2017; ref. 38).

Supplementary Figure S1. Purification, cleavage, and aggregation of GST-mEx1-Q51 with and without GST tag cleavage. **(a)** SDS-PAGE gel stained with Coomassie showing AcTEV cleavage of GST-mEx1-Q51 over time, with 0 h indicating intact protein before addition of AcTEV protease (Invitrogen) and “M” indicating protein ladder (BioRad) with relevant molecular weights listed in kDa. **(b)** ThioflavinT aggregation reaction (top) of GST-mEx1-Q51 including no-AcTEV control (no aggregation) and three technical replicates with AcTEV added (aggregation reaction), and close-up of first five hours of aggregation reaction (bottom), showing minimal aggregation before 2 h (i.e., all aggregation happens post-AcTEV cleavage).

Reviewers' Comments:

Reviewer #2:

Remarks to the Author:

The authors have addressed the reviewers' points in a satisfactory way, and I have no further comments.

Reviewer #3:

Remarks to the Author:

The revised manuscript is substantially improved. The authors have done a reasonable job in addressing my comments and those of the other referees.